# A New and Profitable Protocol to DNA Extraction in *Limnospira maxima*

**DOI:** 10.3390/mps6040062

**Published:** 2023-07-01

**Authors:** Yirlis Yadeth Pineda-Rodriguez, Marcelo F. Pompelli, Alfredo Jarma-Orozco, Novisel Veitía Rodríguez, Luis Alfonso Rodriguez-Paez

**Affiliations:** 1Facultad de Ciencias Agrícolas, Universidad de Córdoba, Montería 230002, Colombia; yadeth@fca.edu.co; 2Laboratorio de Biología Molecular Aplicada (INVEPAR) de la Facultad de Ciencias Agrícolas, Universidad de Córdoba, Montería 230002, Colombia; ajarma@correo.unicordoba.edu.co; 3Instituto de Biotecnología de las Plantas, Universidad Central “Marta Abreu” de Las Villas, Santa Clara 54830, Cuba; novisel@ibp.co.cu

**Keywords:** DNA extraction, A260/A280, PCR, *Arthrospira maxima*

## Abstract

*Limnospira maxima* is a remarkable organism showing great potential as a versatile and sustainable food source, offering a powerful solution to address the pressing issues of malnutrition and undernourishment worldwide. *L. maxima* contains high amounts of proteins, vitamins, minerals, and essential fatty acids. It can be grown in both bioreactors and open systems; however, before considering industrial production, optimization studies of the cultivation must be conducted to obtain knowledge about the ideal environmental conditions. Additionally, for the molecular typing of *L. maxima* strains and their industrial scaling, high-quality and large quantity DNA extraction is required. Notwithstanding, DNA extraction from *L. maxima* can be challenging due to the low amount of DNA in cells and the presence of difficult-to-remove substances such as polysaccharides and polyphenols. In this study, the quality and quantity of DNA extracted from two types of *L. maxima* samples (*Limnospira maxima* strain SISCA accession GenBank: OR195505.1) were evaluated using three commercially available DNA extraction kits and two types of input biological material. The results showed that Pbact-P kit had the highest quantity and quality of DNA, while CTAB-P allowed for a higher quantity and quality of RNA, making them optimal protocols for nucleic acid extraction to improve PCR, rt-PCR, and genome sequencing of *L. maxima* compared with other extraction methods.

## 1. Introduction

*Limnospira maxima* (Setchell & N.L.Gardner) (Nowicka-Krawczyk, Mühlsteinová & Hauer, 2019) [1] (*synon*. *Arthrospira maxima* Setchell & N.L.Gardner, 1917) [2] is an autotrophic filamentous cyanobacterium, formerly incorrectly named blue-green algae. It derives its name from the open left-hand helical or spiral structure of its filaments [3,4]. Its high nutritional content justifies research about its production in high quantities in sufficient volume to permit worldwide commercialization. *L. maxima* contains between 50% and 60% protein [5], and is a good source of vitamins, especially vitamin B12 and carotenoids, of minerals such as iron, and of essential fatty acids such as gamma-linolenic acid [6].

*L. maxima* can be easily cultivated in bioreactors and open, natural, and artificial systems. For its cultivation, a liquid culture medium with salts such as sodium bicarbonate and nitrates is required, which contributes to the alkalinity of the medium, one of the most important factors for scaling up and production.

Before cultivating *L. maxima* on an industrial scale, prior cultivation optimization must be performed in order to gain knowledge about the environmental conditions that favor its growth and cultivation. This is mainly due to the lack of quality controls in the use of efficient and harmless strains. Therefore, the molecular typing of *L. maxima* strains using molecular techniques such as DNA sequencing, DNA amplification by polymerase chain reaction (PCR), and gene expression by rt-PCR, is of great importance for industrial scaling. However, molecular typing requires the extraction of DNA in good quality and quantity, which is quite difficult to extract. Although several methods for extracting DNA from cyanobacteria have been reported, their respective efficiencies can vary widely from one species to another due to the wide morphological, metabolic, and ecological diversities found within each cyanobacterial strain [7]. Cells of cyanobacteria have a very low amount of genomic DNA compared to other bacteria [8]. Nucleic acids (DNA and RNA) represent about 4% of the dry weight of an *L. maxima*, while it can reach up to 20% in fast-growing bacteria such as *Bacillus subtilis* [3]. In addition, the *L. maxima* is rich in substances that are difficult to remove from DNA extracts, such as polysaccharides and polyphenols, and they have a significant number of restriction endonucleases, which represents an additional difficulty for DNA extraction.

The main goal of this study was to evaluate the quality and quantity of DNA extracted from two types of *L. maxima* samples using three DNA extraction methods available on the market.

## 2. Experimental Design

A 3 × 2 factorial design was established with 10 repetitions, using 3 commercial extraction kits, and 2 sample pretreatments as presented in Table 1.

## 3. Procedure

### 3.1. Cyanobacteria Strain, Growth, and Production Medium

The cyanobacteria strain was obtained from the culture collection of the Aquatic Health and Water Quality Laboratory at the University of Cordoba-Colombia, Montería, Colombia (8°47′037″ N; 75°50′51″ W, 15 m a.s.l.), experiencing a mean annual rainfall of 1346 mm, relative humidity of 84%, and mean annual temperature of 27.4 °C [9]. The strain was cultivated in a modified Zarrouk medium [10], with volumes ranging from 1 L to 16 L at the laboratory scale under controlled conditions of temperature (24.5 ± 0.5 °C), photoperiod (12/12), pH (9.5 ± 0.5), sea salt (1 g L^−1^), and constant aeration by an aquarium pump (Sun Sun Submersible Pump, mod. HJ-111, Sunsun, Zhoushan, China) air flow of 6 L h^−1^.

### 3.2. Molecular Identification of the Strain Sampling

Pre-pulverized samples were taken using an APWONE Electric Grinder (Beijing, China). 20.2 mg of the sample were added to sterile and labeled 1.5 mL microvial tubes. Wet samples were taken from 1 mL of a cell suspension. Once collected, the samples were centrifuged at 15,000× *g* for 10 min at 4° (Microcentrifuge mode MicroCL 21R, Thermo Scientific, Missouri, TX, USA) and the precipitate (~1.5 mg) was used for genomic DNA extraction.

### 3.3. Genomic DNA Extraction

Three nucleic acid extraction kits were evaluated, including the Cetyltrimethylammonium Bromide (CTAB 2X) method [11], with modifications (kit 1) as well as two commercial kits following the manufacturer’s instructions (kit 2 and kit 3; Table 1) from the renowned brand (Invitrogen™, Thermo Fisher Scientific, Missouri, TX, USA). For the kit 1, 540 µL of CTAB 2X solution was added to the samples (described in Table 2), followed by 60 µL of 10% polyvinylpyrrolidone (PVP) solution (Merck KGaA, Darmstadt, Germany, part number: P5288). They were then subjected to a water bath for 60 min at 65 °C with vortex shaking every 15 min, and once removed from the bath, they were allowed to stand for 5 min at room temperature. Following this, 900 µL of 24:1 chloroform-isoamyl alcohol solution (Merck KGaA, part number: 25666) was the samples were shaken by inversion for 5 min. After that, they were centrifuged at 15,000× *g* for 5 min at 4 °C. The supernatant was then transferred to a new 1.5 mL reaction tube with 600 µL of isopropanol (Merck KGaA, part number: I9516) and mixed by inversion. The samples were cooled at −20 °C for 10 min, centrifuged for 3 min, and the supernatant was discarded. The resulting pellet was washed with 600 µL of 70% ethanol (Merck KGaA, part number: 65350-M), followed by another step in the centrifuge; the ethanol was discarded and finally, 100 µL of DNA rehydration solution was added. For each extraction methodologies, two types of samples were evaluated: a wet biomass taken from 1mL of cell suspension (~1.5 mg) and pulverized biomass (20.2 mg). Once the DNA was extracted, the quantity and purity were evaluated using 2 µL of the sample in a NanoDrop OneC spectrophotometer (Thermo Scientific, Missouri, TX, USA). The amount of DNA was quantified at 260 nm, while purity was determined by the 260/280 and 260/230 nm absorbance ratios. An A260/A280 ratio of ~1.8 and A260/A230 ratio of 2.0–2.2 indicate DNA “purity”. When the ratio was lower, it indicated the presence of proteins, phenols or other contaminants that are absorbed at these wavelengths [12].

### 3.4. DNA Amplification by Polymerase Chain Reaction (PCR)

Samples with higher concentrations and better quality of DNA were amplified using the PCR technique in a thermocycler (Bio-Rad T100™ thermocycler, Hercules, CA, USA), using the universal primers 27F, 5′-AGAGTTTGATCMTGGCTCAG-3′ and 1492R, 5′-TACGGYTACCTTGTTACGACTT-3′ (Sigma-Aldrich, Darmstadt, Germany, part number 200-00485), which amplify the 16S rRNA gene [13]. The reaction was prepared in a final volume of 50 µL, which contained 25 µL of DreamTaq Hot Start PCR Master Mix (2X) (DreamTaq™ Hot Start PCR Master Mix, Thermo Fisher Scientific, part number K9011), 1 µL of each 10 mM primer, 2 µL of DNA (~157 ng/µL) and sufficient molecular grade water to reach the desired volume. The PCR products were evaluated by electrophoresis on 1% agarose gels in a horizontal electrophoresis chamber and then visually analyzed in a photodocumenter (Enduro GDS Labnet, Tewksbury, MA, USA). The size of the fragments was determined by comparison with a 1 Kb molecular weight marker (Sigma-Aldrich, Darmstadt, Germany, part number D0428).

### 3.5. Sequencing of PCR Products and Sequence Analysis

The PCR products were sequenced using the Sanger technique [14]. The reading of the sequences in AB1 format was performed with the 4Peaks software v 1.8 (Nucleobytes B.V. Gerberastraat 117 1431 RA Aalsmeer, The Netherlands. KvK Amsterdam 56362757), which is linked to the bioinformatic tool Blast, to compare each nucleotide problem sequence (Query Sequence) with those in the GenBank database of the NCBI (United States National Center for Biotechnology Information (http://www.ncbi.nlm.nih.gov—accessed on 2 June 2023)

### 3.6. Statistical Analysis

The data were analyzed using SigmaPlot for Windows v. 14.0 (Systat Software, Inc., San Jose, CA, USA). As necessary, the data were transformed to meet the assumptions of normality and homogeneity under the Kolmogorov–Smirnov test. All media were tested by significance in accordance with Student–Newman–Keuls method (SNK; *p* ≤ 0.05).

## 4. Results

During the study, significant differences were observed in nucleic acid yields and purity ratios among the different types of samples and extraction methods, as well as the interaction between both factors. It was found that treatment T2 yielded the highest nucleic acid concentration at 2.134 ng µL^−1^, followed by T4 at 157 ng µL^−1^. On the other hand, the yields from the T1 had the lowest yield at 20.4 ng µL^−1^, followed by T6 at 12.5 ng µL^−1^, T3 6.9 ng µL^−1^, and T5 at 3.9 ng µL^−1^. The yields from the T3 and T5 treatments were not significantly different but differed from the T1. Furthermore, it was confirmed that the results from the T2, T4, and T6 treatments were also different from each other, as shown in Figure 1A and Table 3. It is important to mention that the minimum accepted DNA concentration is 20 ng µL^−1^.

Regarding the absorbance ratios presented in Figure 1B, it was determined that T4, T5, and T6 treatments had the best A260/A280 purity ratios, with values close to 1.8, which are considered “pure” for DNA. However, the DNA concentrations of T5 and T6 treatments were too low, at 3.9 and 12.5 ng µL^−1^, respectively.

In the case of RNA, a ratio of around 2.0 is considered “pure”. Therefore, sample solutions obtained using the T2 with an A260/280 ratio of 2.2 corresponded to pure RNA. Similarly, absorbance at 230 nm is used as a secondary measure of nucleic acid contamination, and the 260/230 values for “pure” nucleic acid should be between 2.0 and 2.2.

Although the DNA from the T4, T5, and T6 treatments was “pure”, the A260/A230 absorbance ratios were lower than expected ranges, indicating the presence of contaminants that absorb at 230 nm. In contrast, the T1 and T3 treatments do not provide results meeting required purity ranges; therefore, their functionality for DNA extraction cannot be utilized. Instead, the T2 (lane CP) meets the purity ranges for RNA and can be reliably used for RNA extraction during gene expression analyses in this species (Figure 2).

Molecular typing through PCR using universal primers that amplify the 16S ribosomal DNA gene was successful with the T4, corresponding to the PureLink™ Genomic DNA Mini Kit, which is available in the GenBank DNA sequence database of the United States National Center for Biotechnology Information [NCBI] with the accession code GenBank: OR195505.1 [15] (https://www.ncbi.nlm.nih.gov/nuccore/OR195505, accessed at 2 June 2023).

## 5. Discussion

Our results showed that the PureLink™ (Genomic DNA Mini Kit, Invitrogen™, Thermo Fisher Scientific, Missouri, TX, USA), using pulverized biomass (Pbact-P), was the best method for DNA extraction. This kit has been previously used in some studies with cyanobacteria [16,17,18]. However, these studies discovered that when wet biomass was used, the yields were not efficient. This is because *L. maxima* cells are difficult to break down with enzymatic treatments alone, due to their complex multilayered cell wall composed of β-1,2-glucan and peptidoglycan [19]. Therefore, a pre-lysis with grinding is necessary to induce cell lysis. It should also be noted that *L. maxima* cells have a low nucleic acid content, measuring 4% of their dry biomass [3]. The proportion of DNA ranges from 0.6 to 1% [8].

The other treatments were not efficient in terms of yield or purity; therefore, they are not functional for DNA extraction from *L. maxima*. Although the T4 kit allowed for considerable amounts of DNA to be obtained, the absorbance ratios A260/A230 were below the expected values, suggesting the presence of organic compounds. These results are consistent with those obtained by Morin et al. [7], who evaluated five DNA extraction methods for filamentous *L. maxima* and found that the samples obtained using the Fastprep method had significantly compromised purity due to minor protein contamination and the presence of significant amounts of salts and/or phenols that are absorbed at 260 nm. On the other hand, the extracts obtained using the method of Baurain et al. [20] showed significant salt contamination, as indicated by their low 260/230 ratio.

The treatments using the PureLink^®^ Plant Total DNA extraction kit, with both wet and pulverized samples (Pplant-W and Pplant-P), were not effective for extracting DNA from *L. maxima* (DNA concentrations < 20 ng µL^−1^) in our study. However, Faldu et al. [21] successfully isolated DNA from cyanobacteria using this kit.

The CTAB method has been widely used for DNA extraction from plants and fungi due to its ability to remove proteins and polysaccharides present in samples, which can significantly improve the efficiency of the procedure and the quality of the extracted DNA [11]. Our results show that the CTAB 2x kit using pulverized biomass (T2) was not efficient for DNA extraction from *L. maxima*, but was effective for RNA extraction; this might possibly be due to strong DNA breakage by mechanical lysis, preventing the extraction of intact high-molecular-weight DNA. Additionally, *L. maxima* cells have a significant amount of restriction endonucleases [22]. These enzymes protect against the incorporation of external DNA, but also hinder DNA extraction by acting as a self-destruction mechanism during an extraction process.

Although the T1 produced acceptable concentrations of DNA (20.4 ng µL^−1^), the A260/A280 (~1.3) and A260/A230 (~1.7) values also indicate the presence of contaminants in the extracted DNA. This result is consistent with the study by Mak and Ho [23], who reported the extraction of 55 μg of total nucleic acids from *Arthrospira platensis* using 30 mg of wet weight. However, the DNA extracted in this study also showed impurities (260/280 ~1.4), and it has been suggested that the apparently higher yields are due to the presence of RNA and proteins, which can lead to an overestimation of the yields based on absorbance. On the other hand, successfully employed this method for DNA extraction during the identification of a strain of *L. maxima*, but the concentrations obtained were not reported [24].

## 6. Conclusions

The extraction of DNA from the *L. maxima* is a challenging process due to its complex cell wall and low nucleic acid content. In this study, different DNA extraction kits were evaluated, and it was found that the PureLink™ Genomic DNA Mini Kit, using pulverized biomass, provided the most effective method for DNA extraction from *L. maxima.* However, the presence of residual carbohydrates and phenol in the samples extracted with this kit were identified as a problem. Treatments with the PureLink^®^ Plant Total DNA Kit and the CTAB method were also evaluated, but adequate efficiency for DNA extraction from *L. maxima* was not found. Despite these issues, these methods were found to be effective in extracting DNA in other studies related to cyanobacteria. It was evident that wet biomass was not effective for DNA extraction due to the resistance of *L. maxima* cells to lysis, resulting in low DNA yields. Additionally, it was found that the CTAB 2x method using pulverized biomass was not effective for DNA extraction but was effective for RNA extraction from *L. maxima*, possibly due to strong DNA breakage by mechanical lysis and the presence of restriction endonucleases. Overall, this study provides valuable information on nucleic acid extraction from the *L. maxima* and highlights the importance of evaluating different DNA extraction methods for each particular species. Additionally, the presence of contaminants and the quality of the extracted DNA should be considered when interpreting results obtained through spectrophotometric techniques. Finally, DNA extraction from *L. maxima* is a challenging process due to the complexity of its cell wall and low nucleic acid content. Although an effective method for DNA extraction and another for RNA extraction were found in this study, limitations, and considerations for the interpretation of the results obtained must be taken into account.

## Figures and Tables

**Figure 1 mps-06-00062-f001:**
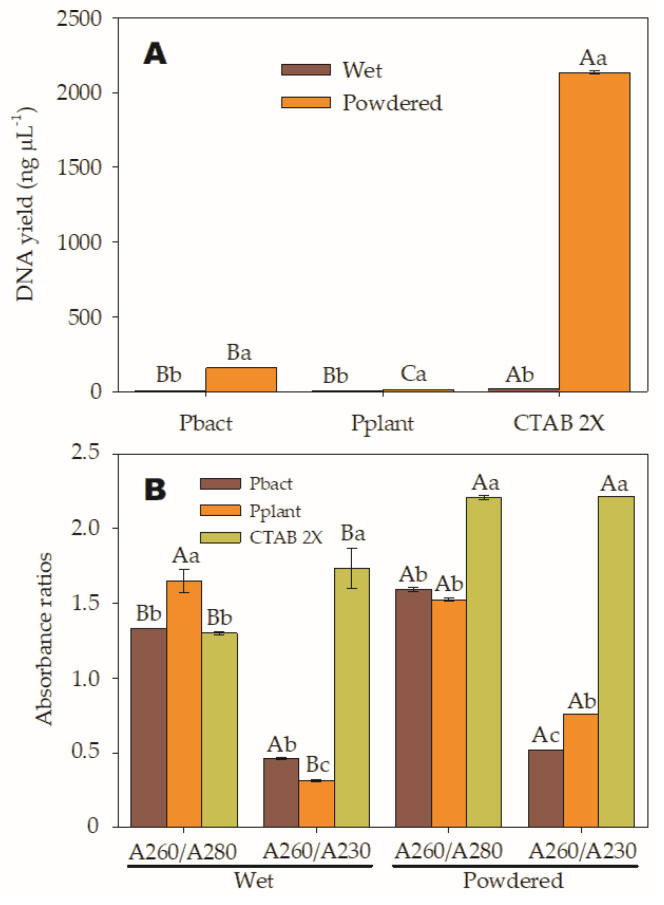
(**A**). Average concentrations of extracted DNA from wet and pulverized *A* samples using two commercial extraction kits: pureLink^®^ Plant Total DNA (Pplant) and PureLink™ Genomic DNA Mini Kit (Pbact) and the modified Cetyltrimethylammonium bromide (CTAB 2X) method are displayed. The lowercase letters indicate the differences between samples (wet and pulverized) evaluated within each kit (SNK *p* < 0.0001); and while uppercase letters compare the mean differences mean between methods with the same type of sample (SNK *p* < 0.0001). (**B**). The purity ratios A260/A280 and A260/A230 were assessed for each sample type (wet and pulverized) and extraction method (Pbact, Pplant, and CTAB 2X). Lowercase letters indicate significant differences among the three extraction methods within each absorbance or purity ratio (A260/A280 and A260/A230) (SNK *p* < 0.0001), while uppercase letters denote significant differences between the two sample types (wet and pulverized) when compared within the same purity ratio (SNK *p* < 0.0001).

**Figure 2 mps-06-00062-f002:**
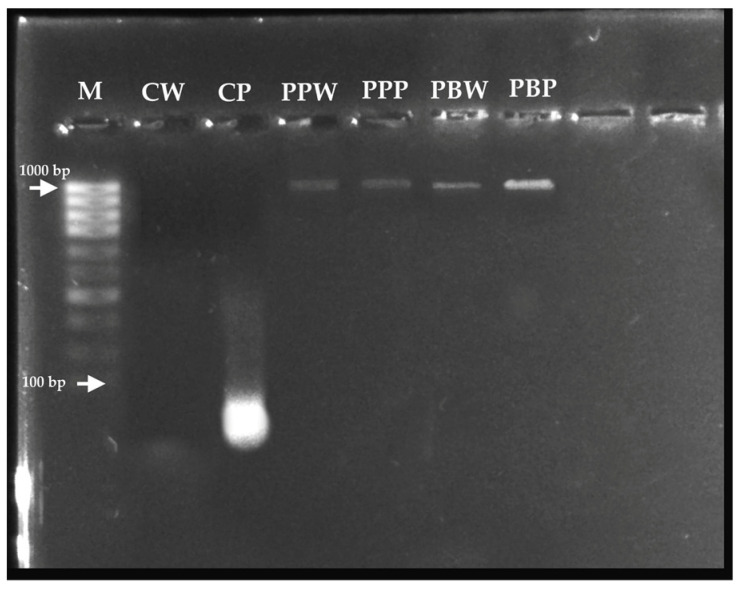
A 1% agarose gel electrophoresis was performed to analyze the nucleic acids extracted from the different treatments under evaluation. M: Molecular weight marker; CW: Ctab-w; CP: Ctab-p; PPW: Pplant-w; PPP: Pplant-p; PBW: Pbact-w and PBP: Pbact-p.

**Table 1 mps-06-00062-t001:** Factorial design 3 × 2.

Extraction Method	Sample Type	
	Wet Sample (W)	Powdered Sample (P)
Cetyltrimethylammonium bromide (CTAB 2X) (Kit 1)	T1. CTAB-W	T2. CTAB-P
PureLink™ Genomic DNA Mini Kit (Pbact) (Kit 2)	T3. Pbact-W	T4. Pbact-P
PureLink^®^ Plant Total DNA (Pplant) (Kit 3)	T5. Pplant-W	T6. Pplant-P

**Table 2 mps-06-00062-t002:** Preparation of the solution (CTAB 2X).

Solution	Final Concentration	10 mL
dH2O (Merck KGaA, part number: 38796)		5.6 mL
1 M Tris-HCl- 7.5 (Merck KGaA, part number: T3253)	100 mM	1.0 mL
5 M NaCl (Merck KGaA, part number: 1064041000)	1400 mM	2.8 mL
0.5 M EDTA-8.0 (Merck KGaA, part number: 324506)	20 mM	0.4 mL
CTAB (Merck KGaA, part number: H6269-500G)	2% (*w*/*v*)	0.2 g
PVP 10% (*w*/*v*) (Merck KGaA, part number: T3253-500G)	1% (*v*/*v*)	0.1 mL

**Table 3 mps-06-00062-t003:** Average values of DNA concentrations and purity ratios in each of the evaluated treatments.

Treatments	Nucleic Acid Concentration (ng/µL)	Purity Ratios	
		A260/A280	A260/A230
T1. CTAB-W	20.4	1.3	1.7
T2. CTAB-P	2134.30	2.2	2.2
T3. Pbact-W	6.9	1.3	0.4
T4. Pbact-P	157	1.6	0.5
T5. Pplant-W	3.9	1.6	0.3
T6. Pplant-P	12.5	1.5	0.7

## Data Availability

The DNA gene was is available in the GenBank DNA NCBI with the at https://www.ncbi.nlm.nih.gov/nuccore/OR195505, accessed on 2 June 2023.

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
