# Peer review of "A New and Profitable Protocol to DNA Extraction in Limnospira maxima"

_mps, 2023, doi:10.3390/mps6040062_

Round 1

Reviewer 1 Report

Line 25: change "commercially DNA extraction kits" to "commercially available DNA extraction kits".

Line 32-33: Referencing style is not consistent with the rest of the paper.

Line 99: Give the full meaning of "PVP": first-time usage.

Line 103:  not clear; "was the samples were shaken".

Line 144: why "Expected Results" and not just "Results"?'

Line 164-165: Figure 1B does not show T4, T5, and T6, modify accordingly.

Line 169-170: No description of the Letters (CW, CP, PPW,......) on the agarose gel.

Line 224: Insert reference; "This result is consistent with the study by , who reported"

Reviewer 2 Report

The manuscript entitled "A new and profitable protocol to DNA extraction in Arthro- 2

spira maxima" deals with an interesting subject that fits nicely into the scope of the journal. I have some suggestion.

-       Figure 1 needs to be redesigned to be more clear and informative (specially Figure 1A).

-       Please add “Arthrospira maxima“ in the Keywords (instead of Limnospira maxima).

-       You can design a Figure for the Procedure section to represent and summarize the steps of your work.

-       There are few weaknesses with respect to grammar, style, and wording that need to be addressed before the manuscript is considered for publication.

Reviewer 3 Report

this is a clean and straightforward method comparison paper for a potentially important process in future industrial applications. The authors describe a simple but robust experimental design to allow efficiency comparisons between the extraction kits, with clear recommendations presented based in the results.

However simple and clear the manuscript is, there is some room for improvement in the communication and presentation of the study, as well as some doubts that arise from the process, as presented below.

Procedure: I think the entire procedure should be presented before statistical analysis, together with the experimental design. This is mostly to better represent the timeline of the study (as statistics happen in the end of the protocol), but I do believe it bring clarity to the procedure.

one question: is it possible that the pre-extraction procedures might have some ill effects on DNA/RNA integrity? have those risks been factored in?

also: what were the reasons for the choice of extraction kits?

Results: the results could benefit greatly from tables presenting the values of the measured purity parameters. while they are described generally in the text, a table with the exact numbers presented for each of the 6 treatments could help placing the effects of each treatment in relation to one another and make clearer links between them.

I would also argue that "expected results" is not an adequate heading for the results section, as there are clear empirical results being presented. If possible, changing the section header to "results" would be recommended.

Round 2

Reviewer 1 Report

The suffix "-W" in Table 1 is not consistent with "-H" in newly introduced Table 3. Rectify.

Author Response

The authors apologize for the error and inform that the suggestions were accepted and corrected in Table 3